

# On the dynamic distinguishability of nodal quasi-particles in overdoped cuprates

**Kamran Behnia**

LPEM (CNRS-Sorbonne University), ESPCI Paris,
PSL University, 75005 Paris, France

## Abstract

$La_{1.67}Sr_{0.33}CuO_4$ is not a superconductor and its resistivity follows a purely $T^2$ temperature dependence at very low temperatures. $La_{1.71}Sr_{0.29}CuO_4$, on the other hand, has a superconducting ground state together with a T-Linear term in its resistivity. The concomitant emergence of these two features below a critical doping is mystifying. Here, I notice that the electron-electron collision rate in the Fermi liquid above the doping threshold is unusually large. The scattering time of nodal quasi-particles expressed in a dimensionless parameter $\zeta$ is very close to what has been found in liquid $^3$He at its melting pressure. In the latter case, fermionic particles become dynamically distinguishable by excess of interaction. Ceasing to be dynamically indistinguishable, nodal electrons will be excluded from the Fermi sea. Such non-degenerate carriers will then scatter the degenerate ones within a phase space growing linearly with temperature.


# 1  Introduction

Elementary particles of a quantum fluid are indistinguishable. Leggett [1,2] argued that it is thanks to this indistinguishibality that such fluids are governed by quantum statistics [and not only quantum mechanics]. Trachenko and Zaconne [3,4] recently highlighted the dynamical aspect of this indistinguishibality and used it as a departing point to explore the boundary between the statistics-active and the statistics-inactive regimes of quantum fluids.

The normal state of cuprate superconductors is nowadays called a 'strange metal'. The expression refers to the puzzling temperature dependence of their electrical resistivity (for recent reviews, see [5] and [6]). The focus of the present paper is a very specific point of the cuprate phase diagram. In hole-doped cuprates, the superconducting dome ends when doping level exceeds a threshold of $p \approx 0.3$. Hussey and collaborators carried out an extensive study of the evolution of resistivity in $La_{1-x}Sr_xCuO_4$ [7]. They found that the superconducting dome and strange metallicity emerge concomitantly when $x < 0.3$. $La_{1.67}Sr_{0.33}CuO_4$ is not superconducting and its resistivity follows $T^2$ [8], but $La_{1.71}Sr_{0.29}CuO_4$ is a superconductor and its resistivity does not correspond to what is expected for a Fermi liquid. Instead, it contains a T-Linear term [7] (Figure 1). This observation is not exclusive to overdoped cuprates. Taillefer pointed out that the normal-state T-linear scattering and the onset of superconducting instability are linked in several other families of superconductors other than cuprates [9]. Greene and collaborators found that in electron-doped $La_{2-x}Ce_xCuO_4$(LCCO), T-square superconductivity emerges only upon the destruction of superconductivity by overdoping [10].

In this paper, I argue that the notion of dynamic distinguishibality [4] illuminates the birth of a 'strange metal' at this locus of the phase diagram. The argument is based on scrutinizing the amplitude of T-square resistivity in the Fermi liquid $La_{1.67}Sr_{0.33}CuO_4$, by comparing it with other Fermi liquids, and by recalling the fate of fermion-fermion collisions when $^3He$ solidifies [11–14].

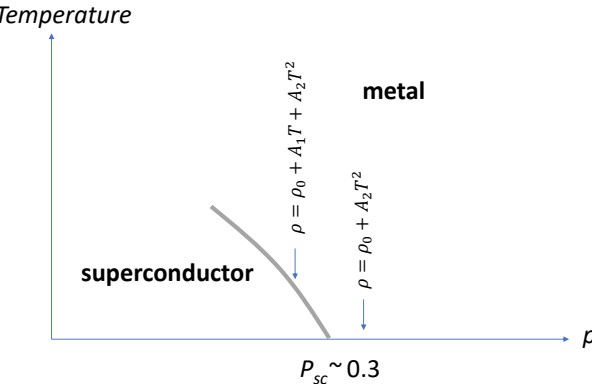

Figure 1: **The puzzle:** Zoom on the cuprate phase diagram near the end of the superconducting dome. Hussey and his co-workers [7] found that below a threshold doping level, the system has a superconducting ground state and a resistivity which follows $\rho = \rho_0 + A_1 T + A_2 T^2$. Above this threshold, the system is not a superconductor and resistivity can be fit with a purely quadratic temperature-dependnet term: $\rho = \rho_0 + A_2 T^2$.

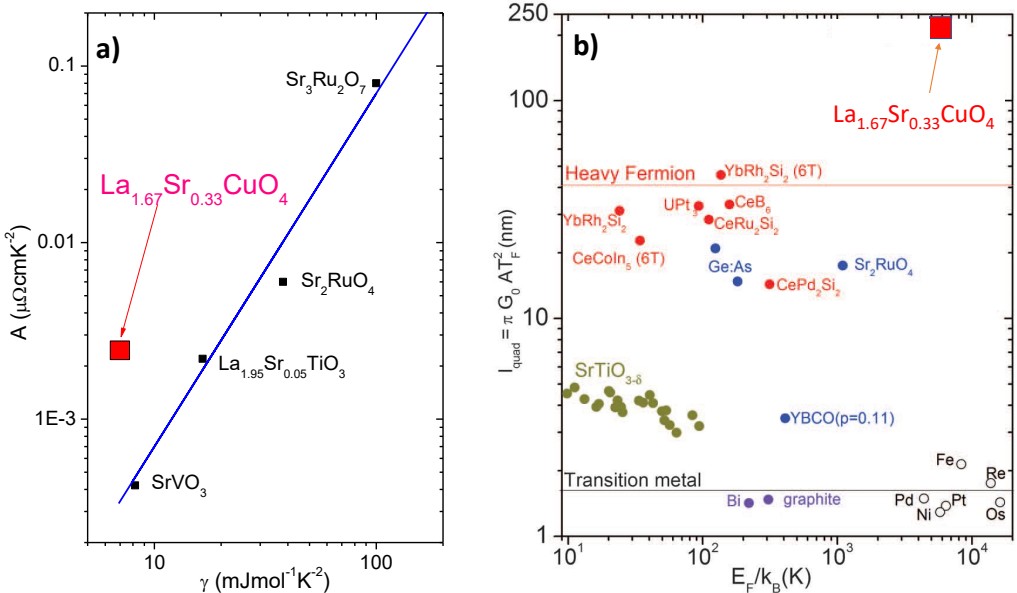

Figure 2: **Standing out among Fermi liquids:** a) The magnitude of the prefactor of T-square resistivity, *A, vs.* the Sommerfeld coefficient, $\gamma$, of several metallic perovskites. $SrVO_3$ [15], $Sr_2RuO_4$ [16], $Sr_3Ru_2O_7$ (at zero magnetic field [17,18]), and $La_{1.95}Sr_{0.05}TiO_3$ [19] all follow Kadowaki-Woods scaling. In contrast, the magnitude of *A* in $La_{1.67}Sr_{0.33}CuO_4$ [7,8] is five times larger than what is expected given its $\gamma$. b) $\ell_{quad}$, derived from the amplitude of *A* and fundamental constants (see text) in different Fermi liquids [20]. The large magnitude of $\ell_{quad}$ in $La_{1.67}Sr_{0.33}CuO_4$ stands out.

## 2 Heavily-doped LSCO stands out among Fermi liquids

$La_{1.67}Sr_{0.33}CuO_4$ is a Fermi liquid, but not a common one. This can be seen by comparing its T-square resistivity with other metallic oxides. In perovskyte family, a variety of instabilities lead to metal-insulator transitions [21] and a metallic ground state is rare.

Let us pick up several exceptions. $SrVO_3$ is a correlated metal with vanadium in $3d^1$ configuration, which remains metallic when Sr is replaced by co-valent Ca [15]. $SrTiO_3$ is a band insulator and $LaTiO_3$ a Mott insulator, but $Sr_{1-x}La_xTiO_3$ alloys are metallic [19]. Specifically, $Sr_{0.05}La_{0.095}TiO_3$ is a dense metal with almost one electron per formula unit [19]. $Sr_2RuO_4$ is an unconventional superconductor with a Fermi liquid normal state above its critical temperature [22]. $Sr_3Ru_2O_7$ has a non-trivial electronic instability at 7.8 T, but is a correlated Fermi liquid at zero magnetic field [17,18]. The feature they all share with $La_{1.67}Sr_{0.33}CuO_4$ is being a dense metallic perovskyte. None of them, however, is a strange metal or become a high-temperature superconductor.

Fig. 2a shows the amplitude of the prefactor of T-square resistivity *A* as a function of the Sommerfeld coefficient (the electronic T-linear specific heat), $\gamma$ in these metals. This Kadowaki-Woods (KW) plot [23] reveals an anomaly. In correlated metals, the prefactor of T-square resistivity scales with the square of $\gamma$ over five orders of magnitude [24]. This scaling is operative when there is roughly one electron per formula unit [25]). As one can see in Fig. 2a, heavily-doped LSCO does not follow the trend observed in other metallic perovskytes. Its T-square resistivity is more than five times larger than it should be, given the magnitude of its $\gamma$. Interestingly, this is also the case of heavily overdoped electron-doped cuprates [1].

---

[1]R.L. Greene, private communication.

The unusually large amplitude of $A$ in $La_{1.67}Sr_{0.33}CuO_4$ betrays itself in a comparison of *all* Fermi liquids. The Kadowaki-Woods scaling can be extended to dilute metals by plotting $A$ as a function of the Fermi energy, $E_F$ [20, 25, 26]. Due to Pauli exclusion principle, the phase space of scattering among fermions is proportional to $(\frac{k_B T}{E_F})^2$. Dimensional considerations imply [20]:

$$A = \frac{\hbar}{e^2}(\frac{k_B}{E_F})^2 \ell_{quad}.$$

(1)

Here $\hbar$ is the reduced Planck constant and $e$ is the fundamental charge. $\ell_{quad}$ is a phenomenological material-dependent length scale. A survey of available data shows that for all known Fermi liquids $\ell_{quad}$ is between 1 to 50 nm [20, 26]. As one can see in Fig. 2b, in $La_{1.67}Sr_{0.33}CuO_4$, $\ell_{quad} \approx 240$ nm. Decidedly, this Fermi liquid is not a banal one. The unusually large $A$ of this metal, given its density of states and its degeneracy temperature is the first step for understanding its transformation to a strange metal upon the removal of dopants.

To see the significance of this, let us consider the case of normal liquid $^3$He.

## 3  Distinguishibality on the verge of solidification in $^3$He

Under their own vapor pressure, the two isotopes of helium do not solidify down to zero temperature. These quantum liquids [1, 2] become quantum solids upon the application of pressure. With an odd number of protons, neutrons and electrons, a $^3$He atom is a composite fermion. The molar volume changes from 36.84 cm$^3$/mol at zero pressure to 25.5 cm$^3$/mol at p=3.4 MPa, when it solidifies. This is twice larger than what is classically expected (12 cm$^3$/mol) and is a consequence of the large zero-point motion of the atoms in the crystal [14].

The temperature dependence of thermal conductivity and viscosity in normal liquid $^3$He at different pressures have been carefully measured by several authors. The quasi-particle scattering time extracted from these studies, broadly consistent with each other, were reviewed in detail by Dobbs [14]. The scattering time derived from thermal conductivity, $\tau_\kappa$ displays a $T^{-2}$ behavior in the zero-temperature limit [11, 13], as expected for a Fermi liquid.

With increasing pressure, interaction between the atoms, consisting principally of a strong hard-core repulsion and a weak van der Waals attraction, intensifies. This leads to an amplification of the effective mass, quantified by measurements of specific heat [12] (Fig. 3a). Fig. 3b shows the evolution of $\tau_\kappa T^2$ (in the zero-temperature limit) as a function of pressure reported by Wheatley [11] and by Greywall [13]. The trend is similar, but Greywall's values are about 25 percent lower than Wheatley's. The highest pressure (3.44 MPa) corresponds to the onset of solidification. By this pressure, the time between two fermion-fermion collisions has decreased by a factor of almost 3.

Several theoretical studies have examined the evolution of the effective mass and the Landau parameters by pressure. Vollhardt, Wölfle and Anderson [27] employed a Hubbard lattice-gas model with a variable density of particles to describe the pressure dependence of thermodynamic properties of normal liquid $^3$He. Pfitzner and Wölfle [28, 29] gave a reasonable account of transport coefficients under pressure. According to these studies, normal liquid $^3$He is a strongly interacting and almost localized Fermi liquid [30].

What will be scrutinized here are the amplitudes of $\tau_\kappa T^2$ and the Fermi energy, $E_F$ on the verge of solidification. According to Vollhardt and Wölfle [31], the quasi-particle lifetime on the Fermi surface is given by:

$$\tau_N^0 = \frac{64}{\pi^3}\frac{\hbar E_F}{(k_B T)^2}\langle W\rangle_a^{-1},$$

(2)

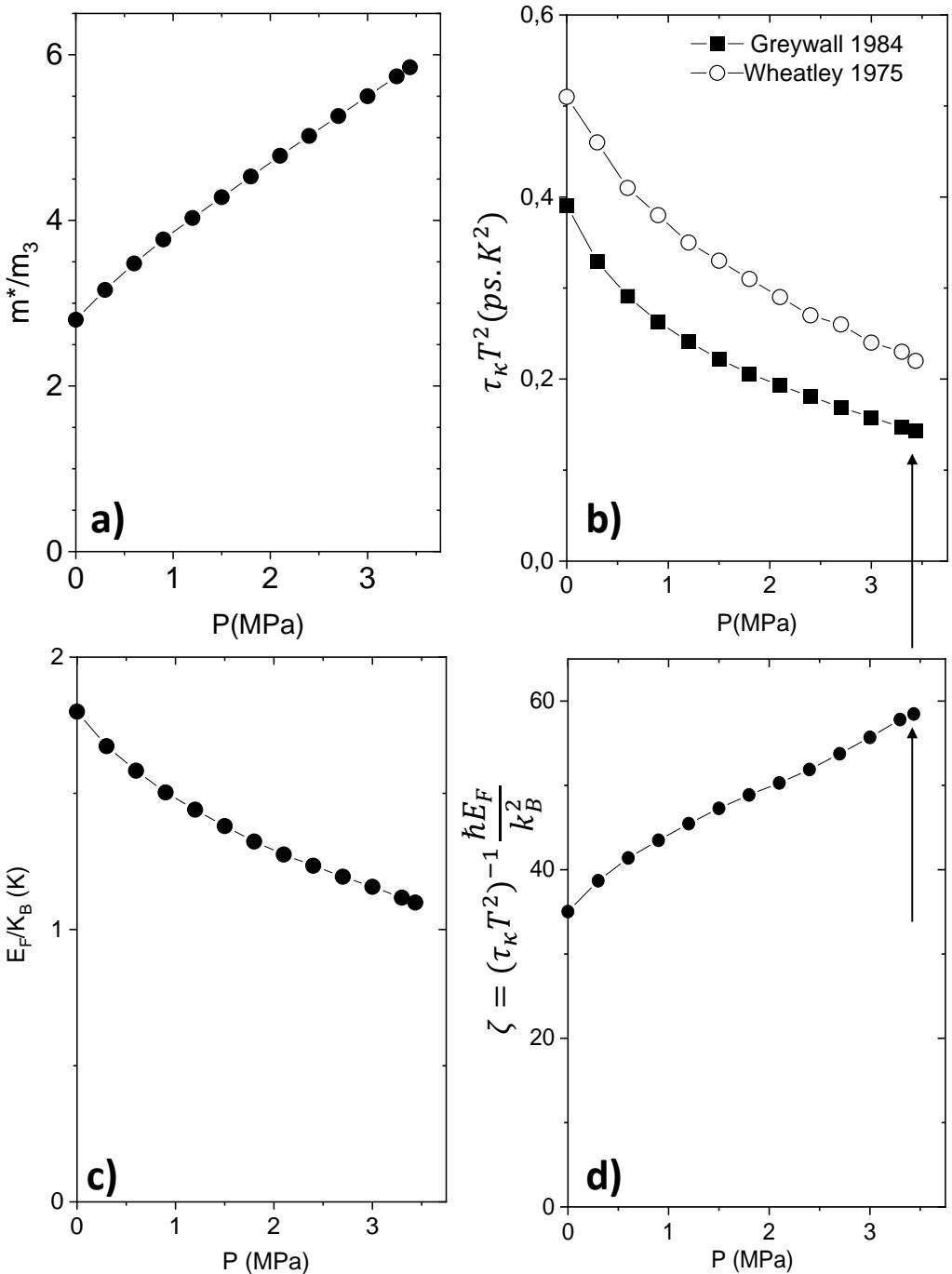

Figure 3: **The case of normal liquid $^3$He :** a) The pressure dependence of the effective mass in $^3$He quantified by measuring the T-linear specific heat [12]. b) The quasi-particle scattering time extracted from thermal conductivity multiplied by $T^2$ as a function of pressure, according to Wheatley [11] and Greywall [13]. c) The pressure dependence of the Fermi energy using the Fermi momentum and the Fermi velocity given by Greywall [13]. d) The pressure dependence of the inverse of the normalised quasiparticle lifetime using Greywall's data (see text). Black arrows show the pressure at which solidification occurs.

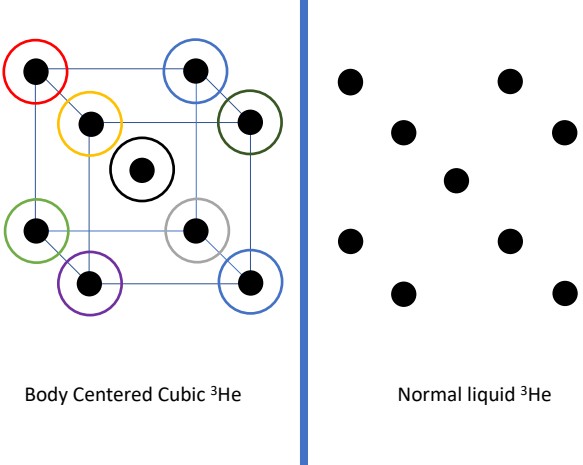

Body Centered Cubic ³He  Normal liquid ³He

Figure 4: **Discernibility and indiscernibility in ³He :** In liquid ³He atoms are indistinguishable, but in solid ³He, they are confined to specific sites in real space. This makes them distinguishable. In the solid near the melting transition, atoms wander around, thanks to their zero-point motion and can exchange their places with their neighbors. On the other hand, in the liquid, collisions in real space confine atoms to a restricted neighbourhood. The liquid solidifies when collisions confine each atom to a spatial neighborhood.

$\langle W \rangle_a$ is the angular average of the transition probabilities between spin singlet and spin triplet states [14,31,32]. Multiplying $\tau_\kappa T^2$ by $\frac{k_B^2}{\hbar E_F}$ will yield a dimensionless number inversely proportional to the fermion-fermion collision strength.

Fig. 3c shows the evolution of the Fermi energy with pressure reconstructed from Greywall's data [13]. Combining this with $\tau_\kappa T^2$ leads to Fig. 3d, which shows the pressure dependence of the normalized scattering time:

$$\zeta = \frac{\hbar E_F}{\tau_\kappa (k_B T)^2} \, . \tag{3}$$

Solidification occurs when this number becomes as large as 60. This implies a huge collision rate between quasi-particles at the verge of solidification. Inserting this number in Eq. 2 leads to the conclusion that at the onset of solidification $\langle W \rangle_a \approx 140$. Surprisingly, this remarkably large value has not been hitherto explicitly noticed, let alone commented.

The value of $\langle W \rangle_a$ (or $\zeta$) is set by the magnitude of dimensionless Landau parameters of the Fermi liquid, which are denoted by $F_l^s$ (spin symmetric) and $F_l^a$ (spin antisymmetric) [29,31]. Practically, Landau parameters with $l < 2$ are the ones which matter and the higher order ones may be safely neglected [31,33]. Vollhardt and Wölfle [31] have used experimental data to calculate the evolution of Landau parameters with pressure and have found that at the threshold of solidification ($P = 3.4MPa$) $F_0^s = 88.47$, $F_0^a = -0.753$, and $F_1^s = 14.56$. These are large numbers. The Fermi liquid picture still holds, albeit restricted to a very narrow temperature, at the onset of solidification and its Landau parameters are large, yet finite.

Let us first note that in standard theories of Mott transition [34], Landau parameters diverge. Let us also note that the mean-free-path of ³He atoms remains much longer than their wavelength even on the verge of solidification . According to Greywall's data, at P=3.4MPa and T=0.01 K, $\tau_\kappa = 1.43ns$, $v_F = 32.4m/s$, $\ell = 42.6nm$ and $k_F = 8.9nm^{-1}$. Since $k_F \ell \gg 1$, Anderson localization is *not* what drives confinement in space.

I conjecture that what drives solidification is the impossibility for Landau parameters to become arbitrarily large. An infinite $F_0^s$, for example, would make the liquid incompressible,

which is implausible. Vollhardt and Wölfle [31] highlighted the fact that normal liquid $^3$He is much less compressible than a non-interacting Fermi liquid, thanks to its large $F_0^s$. On the other hand, this interacting liquid is not more incompressible than solid $^3$He. Indeed, according to the experiment [35], at the melting pressure, the compressibility of the two phases are nearly equal. Now, the compressibility of solid is set by its phonon spectrum and, one may suspect, this is what sets the bound to $F_0^s$ in the liquid. After all, Landau parameters are two-particle correlators [34] and atoms of a quantum liquid have a finite coordination number [36]. Therefore, two-particle correlations in real space cannot attain an arbitrarily large magnitude.

Solid $^3$He is not subject to Fermi-Dirac statistics, because each atom is confined to the neighborhood of a designated site [37]. Near the melting pressure, in the solid state, neighbouring atoms can frequently exchange theirs sites thanks to their zero-point motion. Each atom frequently encounters its first neighbours far from its lattice site [38, 39], but there is still a one-to-one correspondence between an atom and its designated site. In contrast to the solid, the atoms of the liquid are indistinguishable (See Fig. 4). Collisions allow an atom in the liquid state to 'observe' its neighbours in real space [40]. As these collisions multiply, the atom cannot avoid being confined to a specific and distinguishable volume of the real space.

The case of $^3$He illustrates that the time between two successive fermion-fermion collisions can become unbearably short for the survival of a Fermi liquid, despite a long mean-free-path and $k_F \ell \gg 1$. Let us now return to cuprates.

## 4   Nodal quasiparticles: the unbearable shortness of being

What we saw in the case of $^3$He raises a question: Given the unusually large amplitude of the T-square resistivity in heavily doped LSCO, how short is the normalized quasi-particle lifetime?

Fig. 5a shows the Fermi surface of of La$_{1.68}$Sr$_{0.32}$CuO$_4$ according to a tight binding model with nearest-neighbor hopping parameters chosen to fit the Fermi surface seen by Angle-Resolved Photoemission Spectroscopy (ARPES). [41–43]. The radius of this Fermi surface has a modest angular variation. In comparison, the angular variation of the Fermi velocity, $v_F$, is much larger (see Fig. 5b). This is a consequence of the proximity of Fermi surface and the Brillouin zone boundary along the anti-nodal direction. Because of the van hove singularity, the evolution of the Fermi surface with doping differs along nodal and anti-nodal orientations. As a consequence, the derivative of Fermi energy in momentum space ($v_F$) has a strong angular dependence.

Solving the Boltzmann equation, one finds a general expression for electric conductivity [44]:

$$\sigma = \frac{1}{4\pi^3}\frac{e^2}{\hbar}\int \tau v_k \frac{v_k}{v_k} dS_F . \tag{4}$$

In the case of cubic symmetry, $\sigma$ is identical for the whole solid angle and therefore:

$$\sigma_{cub.} = \frac{1}{3\pi^2}\frac{e^2}{\hbar}\tau v_F k_F^2 . \tag{5}$$

Note that $\tau$, $v_F$ and $k_F$ can be anisotropic. However, cubic symmetry, by constraining $\sigma_{cub.}$ to be isotropic, restricts possible profiles for $\tau(\theta,\phi)$, $v_F(\theta,\phi)$ and $k_F(\theta,\phi)$.

Heavily overdoped LSCO has a tetragonal symmetry and a quasi-cylindrical Fermi surface extending vertically along the whole Brillouin zone. In this case, the in-plane electrical conductivity is constrained to be isotropic and equal to :

$$\sigma_{tet.} = \frac{1}{2\pi c}\frac{e^2}{\hbar}\tau v_F k_F . \tag{6}$$

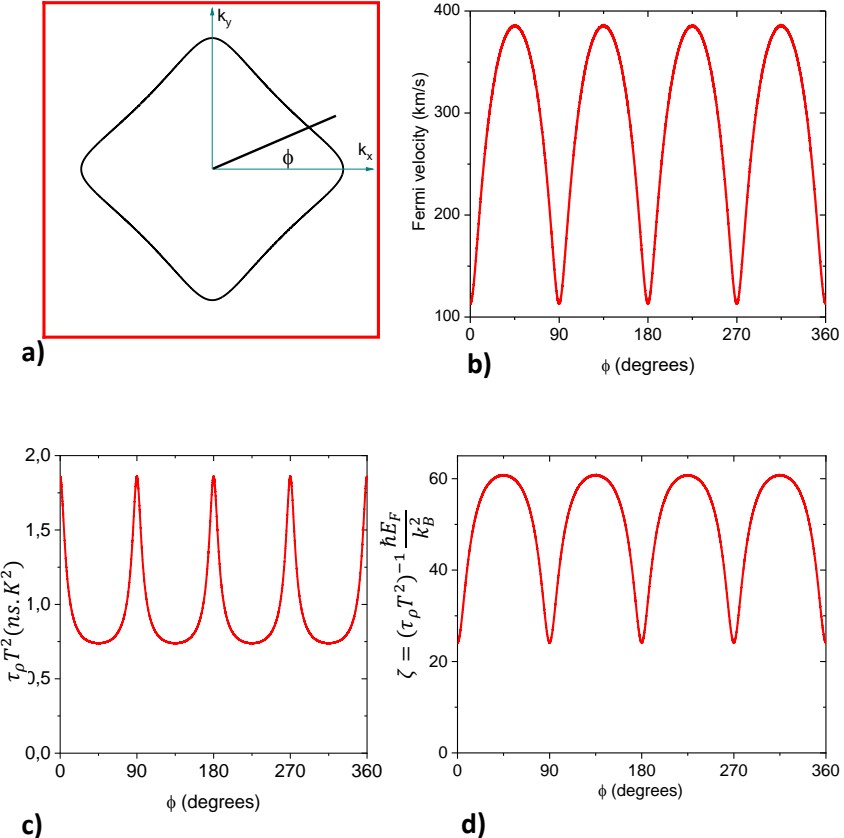

Figure 5: **Angular variation of collision time in heavily overdoped LSCO:** a) Fermi surface and the Brillouin zone of $La_{1.68}Sr_{0.32}CuO_4$; b) Angular variation of the Fermi velocity; c) Angular variation of the scattering time derived from resistivity times $T^2$; d) Angular dependence of dimensionless $\zeta$ extracted from $\tau_\rho T^2$ and $\frac{k_B^2}{\hbar E_F}$. Compare the absolute value of the maximum for nodal orientations with what was seen in the case of $^3$He on the verge of solidification.

Here, $c$ is the lattice parameter along the c-axis. The experimentally measured prefactor of in-plane T-square resistivity ($A = 2.5n\Omega cmK^{-2}$ [7,8]) is indeed isotropic in the basal plane. Using Eq. 6 and $AT^2 \equiv \sigma_{ee}^{-1}$, to quantify $\tau_\rho$ (the index refers to the fact that the experimental probe used to extract this time scale is electrical resistivity):

$$\tau_\rho(\theta)T^2 = \frac{\hbar}{e^2}\frac{2\pi c}{Ak_F(\theta)v_F(\theta)}\,. \tag{7}$$

The Fermi velocity, $v_F$, the Fermi wave-vector, $k_F$ and $\tau_\rho$ have all their angular dependence. Since the anisotropies of $k_F$ and $v_F$ do not cancel out, one expects a significant angle dependence of $\tau_\rho T^2$. As seen in Fig. 5c, this is indeed the case. It is more than twice smaller along the nodal orientation. Note that, because of the large anisotropy of the Fermi velocity, the anisotropy of the mean-free-path is the inverse of the anisotropy of the scattering time. Nodal quasi-particles have a longer mean-free-path yet a shorter scattering time.

Combining $\tau_\rho T^2$ and the Fermi energy $E_F = 5900K$ (extracted from the magnitude of the T-linear electronic specific heat $\gamma$=6.9 mJ.mol$^{-1}$.K$^{-2}$ [8]) leads to the quantification of the dimensionless $\zeta$ and its angular dependence (Fig. 5d). Along the nodal orientation, it becomes as large as $\approx 60$, close to what was found above in $^3$He on the verge of solidification.

Table 1: $^3$He and heavily-doped LSCO compared to heavy-fermion UPt$_3$ [45–47], correlated oxide Sr$_2$RuO$_4$ [22], semi-metallic antimony [48,49] and dilute metallic strontium titanate [20]. Note the exceedingly large normalized amplitude of $\zeta$ in LSCO. UPt$_3$ and Sr$_2$RuO$_4$ despite their larger mass enhancements have smaller $\zeta$s.

| System | $k_F(nm^{-1})$ | m$^\star$/m$_0$ | $E_F$ (K) | $\zeta$ |
|---|---|---|---|---|
| $^3$He (p=0) | 7.9 | 2.8 | 1.8 | 35 |
| $^3$He (p=3.4MPa) | 8.9 | 5.8 | 1.1 | 60 |
| La$_{1.67}$Sr$_{0.33}$CuO$_4$ | 5.6 | 5 | 5900 | 24-61 |
| UPt$_3$ | 5 | 16-130 | 90 | $\approx 10$ |
| Sr$_2$RuO$_4$ | 5 | 3.3-16 | 1800 | $\approx 16$ |
| Sb | 0.8 | 0.07-1 | 1100 | $\approx 0.1$ |
| SrTiO$_{3-\delta}$ (n=4×10$^{17}cm^{-3}$) | 0.23 | 1.8 | 18 | $\approx 0.1$ |

Let us recall that the microscopic interaction between fermions are very different in the two cases. In metals, point-like electrons repulse each other through screened Coulomb repulsion. In contrast, neutral $^3$He atoms interact over a short range comparable to their hard-sphere radius and the interaction has both attractive and repulsive components. Nevertheless, the strength of fermion-fermion scattering rate ($\tau^{-1}$) can be quantified in both cases by the dimensionless $\zeta$.

Only when there is a single Fermi surface and a unique Fermi temperature, the amplitude of $\zeta$ is unambiguous. This is the case of heavily overdoped LSCO, but not other Fermi liquids. Most often, they have multiple anisotropic Fermi pockets. Assuming a single average Fermi energy, one can directly extract from the T-square prefactor, the phenomenological length scale $\ell_{quad}$, introduced first in ref. [20] and defined by Eq. 1, which is proportional to $\zeta$ through a dimension-dependent length scale.

In a simplified one-band approximation, the total carrier density and the measured $T$-linear specific heat yield an average Fermi energy for a given system. In that case, for a three-dimensional metal with a spherical Fermi surface, one has:

$$\zeta_{3d} = \frac{2}{3\pi^2} \frac{e^2}{\hbar} \frac{A}{k_B^2} E_F^2 k_F \,. \tag{8}$$

For a two-dimensional metal with a cylindrical Fermi surface, the expression becomes:

$$\zeta_{2d} = \frac{1}{3\pi c} \frac{e^2}{\hbar} \frac{A}{k_B^2} E_F^2 \,. \tag{9}$$

These expressions, which neglect anisotropy and multiplicity of Fermi pockets, can be cautiously used to extract the rough magnitude of $\zeta$ in various metals.

## 5   Comparison with other metals

Table 1 compares $^3$He and in heavily-doped LSCO with two weakly-interacting and two strongly-interacting Fermi liquids. The list includes antimony [49], dilute oxygen-reduced strontium titanate [50], the heavy-fermion metal UPt$_3$ [45] and the correlated oxide Sr$_2$RuO$_4$ [22].

UPt$_3$ displays a quadratic resistivity below 1.5 K with A=1.55$\mu\Omega$ cm K$^{-2}$ for in-plane charge flow [45]. It has five different Fermi surface pockets with different sizes and effective masses [45–47]. The largest and the most relevant sheet of the Fermi surface is the band 3 ('Oysters

and urchins' [45]) giving rise to the $\omega$ orbit seen by quantum oscillations [47]. Table 1 uses the effective mass and the average radius of this sheet for rough estimates of $k_F$ and $E_F$. The Fermi energy of 90 K is compatible with an alternative estimation using the magnitude of the Sommerfeld coefficient ($\gamma = 450 J/K^2.mol$) and carrier density ($1.4 \times 10^{28} m^{-3}$) in UPt$_3$.

Resistivity in Sr$_2$RuO$_4$ is quadratic below 25 K with A=6 n$\Omega$ cm K$^{-2}$ for in-plane charge flow [16]. Its Fermi surface consists of three warped cylinders [22] with radii ranging from 3.04 to 7.53 nm$^{-1}$ and the effective mass from 3.3 to 16 bare electron masses [22]. The table uses average values for $k_F$ and $E_F$ for Sr$_2$RuO$_4$.

Antimony (Sb) displays a quadratic resitivity below 10 K with a prefactor of 0.8 n$\Omega$ cm K$^{-2}$ [49]. Its Fermi surface has three electron pockets and a single interconnected Fermi surface. The Fermi wave-vector along different orientations varies by one order of magnitude [48]. The table gives an average value for $k_F$ in antimony. The Fermi energy is almost the same for electrons and holes.

SrTiO$_3$ is a band insulator. It becomes a dilute metal with a very low carrier density when a small amount of oxygen atoms are removed [51]. The resistivity of this dilute metal follows a T-square resistivity with A=9 $\mu\Omega$ cm K$^{-2}$ when n=4$\times 10^{17} cm^{-3}$ [20]. At this carrier density, only a single band is occupied and the Fermi surface seen by quantum oscillations can be roughly approximated to a spherical one.

The experimentally resolved $A$ in set by the overall contribution of the sheets of a multi-component Fermi surface. Therefore, the average values yield only a rough estimate of $\zeta$. It can have different values for different pockets. As seen in the table, according to this rough estimate, $\zeta > 1$ in strongly-correlated systems and $\zeta < 1$ in the weakly correlated ones. The maximum $\zeta$ is lower in Sr$_2$RuO$_4$ and in UPt$_3$ than in LSCO (or in $^3$He), despite their larger mass enhancement. Note also that while the magnitude of $A$ in SrTiO$_{3-\delta}$ and in Sb differ by four orders of magnitude, their $\zeta$ is roughly similar.

Thus, $\zeta$ is unusually large in heavily overdoped LSCO. Given the angular variation of $\zeta$, an eventual upper boundary will be encountered by nodal quasi-particles before other electrons of the Fermi sea. Let us assume that an exceedingly large fermion-fermion collision rate freezes the nodal quasi-particles out of the Fermi sea.

# 6   Consequences of nodal freeze-out

What happens to the nodal quasi-particles once are excluded from the Fermi sea is not known. However, let us assume that this happens below x=0.3 and generates two categories of electrons [2]: Those for which quantum statistics is non-operative and others remaining in the Fermi sea. Such a hypothesis will provide new possible solutions to a number of longstanding puzzles. They are listed below:

- **Planckian dissipation:** Bruin and co-workers noticed that in many metals with a T-linear resistivity, the amplitude of scattering time is of the order of $\tau_P = \frac{\hbar}{k_B T}$ [53]. Legros and co-workers [54] have reported that this is indeed the case of overdoped cuprates. This so-called 'Planckian dissipation' is encountered in a variety of contexts in both conventional and unconventional metals [55]. A scattering time of the order of $\sim \frac{\hbar}{k_B T}$ is expected when degenerate electrons are scattered off classical objects. Two examples are such scattering centers are phonons above their Debye temperature or electrons above their Fermi temperature.

  The presence of classical electrons can account for the strange metallicity of Sr$_3$Ru$_2$O$_7$

---

[2]For an alternative scenario referring to two distinct (coherent and incoherent) charge sectors see [52]. Note the very different nature of electron dichotomy in the two scenarios.

[56]. Mousatov and co-workers showed that in this system, $T$-linear resistivity and its Planckian prefactor [53] can be explained by invoking the scattering of degenerate electrons in a large pocket by classical electrons in a small pocket. Such a scenario, which may be relevant to other correlated metals, does not directly apply to cuprates, which have a single Fermi pocket. On the other hand, if nodal quasi-particles become classical, i.e. if they get excluded from the Fermi sea forming a distinct liquid with a distinct degeneracy temperature, then a scenario similar to the one invoked for $Sr_3Ru_2O_7$ [56] can work for cuprates.

- **Isotropic $T^{-1}$ scattering rate:** Grissonnanche and co-workers [57] have recently reported that the T-linear scattering is independent of direction. Let us consider the angular dependence of the $T$-linear scattering time when degenerate electrons are scattered by non-degenerate nodal electrons. In this case, the angular distribution of scattering centers peaks along two orthogonal nodal orientations and this will dictate the angular dependence of the scattering rate. Assuming a square cosine variation for each nodal orientation, since $cos^2(\phi) + cos^2(\phi + \pi/2)$ does not vary with $\theta$, one finds a flat scattering rate.

- **Saturation of the amplitude of the T-square prefactor:** Cooper and co-workers [7] found that the amplitude of the prefactor of T-square resistivity does not increase with doping when it falls below the threshold of strange metallicity. This behavior contrasts with what has been seen in quantum-critical metals [58,59], where $T$-square resistivity diverges to a $T$-linear one. On the other hand, it is compatible with a ceiling for electron-electron scattering met by a subset of electrons.

- **The evolution of carrier density with doping:** Experiments [60,61] have found that the Hall carrier density gradually decreases from $1+p$ at high doping to $p$ at low doping. On the other hand, the superfluid density does not show any sharp feature as a function of doping and shows a dome-like structure similar to the critical temperature [62], as found in a conventional superconductor [51]. In the present picture, with decreasing carrier density, a larger fraction of electrons is peeled off the Fermi surface. On the other hand, the superfluid density, which keeps to be zero along the nodal direction is not affected by this peeling. If the nodal electrons cease to participate in charge transport in the normal state, the discrepancy between the doping dependencies of the normal-state density and the superfluid density may find an explanation.

- **Nodal excitons:** Being extracted off the Fermi sea, nodal electrons and nodal holes can pair up and form excitons, plausible candidates for playing the role of pair-forming Bosons. Such a mechanism for the formation of pairs has been already proposed in other contexts [63], but not in cuprates. Since the nodal electrons do not participate in the Fermi sea, the superconducting order parameter would therefore vanish along the nodal orientations, in conformity with the d-wave symmetry of cuprates [64]. If this happens to be the case, then the the charge order [65] competing with superconductivity is also eventually the one driving the superconducting instability through its fluctuations.

## 7 Concluding remarks

The present paper reports on two observations and on a speculation.

The observations are about the amplitude of fermion-fermion scattering in two different strongly correlated fermioinc systems: $^3$He atoms near the melting pressure and nodal quasi-particles in cuprates on the verge of superconductivity. Their dimensionless amplitude

is strikingly similar and fermion-fermion collision rate in liquid $^3$He is near the threshold of distinguishability.

The speculation is that the large collision rate in the cuprate would render some carriers distinguishable. The exclusion of this subset of fermions from the Fermi can lead to plausible explanations for the sudden emergence of T-linear resistivity and a robust superconducting ground state.

The present approach may also prove relevant to deciphering the passage from T-square to T-linear resistivity in magic-angle twisted bilayer graphene [66], where T-linear and T-square resistivity emerge in close proximity of each other.

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
