# Peer review of "On the dynamic distinguishibality of nodal quasi-particles in overdoped cuprates"

_SciPost Physics, doi:SciPost Phys. 12, 200 (2022)_

## Round 1 · Referee Report · Anonymous (Referee 2) · 2022-5-27

Report

I am pleased to see that the author has responded to most of the comments made by the referees and I am happy to approve publication of the present manuscript in SciPost. Ref. [47] appears to be incorrect - it should be PRL, vol. 76, 3786 (1996).

---

## Round 1 · Referee Report · Anonymous (Referee 3) · 2022-6-5

Report

This paper can now be published. The author has made improvements in the revised version. In my opinion, the points raised were not all fully addressed and there are aspects of the paper that remain obscure to me. These include the treatment of anisotropy, the connection between quadratic and linear temperature dependence of the resistivity, the connection between scattering and indistinguishability, and the precise way the scenario is being realized in cuprates. Nonetheless, the paper is insightful and stimulating and is unlikely to be improved by further revision. Therefore, I recommend publication in its present form.

---

## Round 1 · Referee Report · Anonymous (Referee 1) · 2022-6-6

Report

I agree with the other referees that the paper can be published now. Like them, I do not agree with all its content, but, like them, I think that publishing it will stimulate thought and discussion. It is good that it be seen along with our (referee) scepticism about certain points. I would recommend the reader to look at the original reports and the replies the author has made to them. It will be seen that some of the points raised have been adequately addressed, while others have not. However, the author has tried to respond as well as he could, and I agree with the point made by another referee that the paper is unlikely to be further improved by another round of revisions.

---

## Round 1 · Author Response

I thank all three referees for their helpful inputs and constructive criticisms. In particular, I appreciate the time they have devoted to reading and evaluating the manuscript. The new version has greatly benefited from their comments.

My main observation is that the amplitude of fermion-fermion scattering expressed in dimensionless units is strikingly similar in cuprates and in helium three. This point has not been contested by any of the referees. On the other hand, two points appear to be controversial and need clarification.

1- The role of dynamic distinguishability in solidification of 3He: Admittedly, the previous version of the paper was not sufficiently clear. The new version tries to do a better job by arguing that the rate of collisions in a Fermi liquid cannot become arbitrarily large and this is what drives solidification in 3He. Under pressure, fermions in 3He end up becoming distinguishable because collisions confine them to a specific volume in the real space. Moreover, this is NOT a case of Anderson Localization, justifying the comparison with cuprates.

2- The anisotropy of the mean-free-path in cuprates: I have tried to clarify a misconception. I do not assume that the mean-free-path is isotropic. What is isotropic is the in-plane conductivity. It is constrained to be so by the tetragonal symmetry of the lattice. The mean-free-path is indeed anisotropic and longer for nodal quasi-particles.

I have followed referee 3's recommendation and included an extended discussion comparing zeta in cuprates with zeta in other Fermi liquids.

---

## Round 1 · List of Changes

i) The abstract has changed to enhance clarity.
ii) Section III has been extended in other to clarify the utility of the concept of distinguishability when discussing the solidification of 3He.
iii) Fig. 4 has changed.
iv) Fig. 3d and Fig. 5d now show the evolution of zeta and not its inverse.
v) Section IV has been entirely rewritten to clarify different anisotropies.
vi) Section V is totally new.
vii) Table 1 has been extended.
viii)Many references have been added.

---

## Editorial Decision

published